# Assessment of Executive Functions in Children with Sensorineural Hearing Loss and in Children with Specific Language Impairment: Preliminary Reports

**DOI:** 10.3390/brainsci14050491

**Published:** 2024-05-13

**Authors:** Maria Lauriello, Giulia Mazzotta, Antonella Mattei, Ilaria Mulieri, Alessandra Fioretti, Enzo Iacomino, Alberto Eibenstein

**Affiliations:** 1Department of Biotechnological and Applied Clinical Sciences, University of L’Aquila, 67100 L’Aquila, Italy; lauriellomaria@gmail.com (M.L.); alberto.eibenstein@univaq.it (A.E.); 2Centro di Audiofonologopedia, 00199 Roma, Italy; logopedista.giuliamazzotta@gmail.com (G.M.); ilariamulieri@hotmail.com (I.M.); 3Department of Life, Health and Environmental Sciences, University of L’Aquila, 67100 L’Aquila, Italy; antonella.mattei@univaq.it; 4European Hospital, 00149 Rome, Italy; 5Department of Otolaryngology, San Salvatore Hospital, 67100 L’Aquila, Italy; enzoiacomino74@hotmail.com

**Keywords:** executive functions, sensorineural hearing loss, specific language impairment

## Abstract

Executive functions (EFs) are related abilities, associated with the frontal lobes functions, that allow individuals to modify behavioral patterns when they become unsatisfactory. The aim of this study was to assess EFs in children with sensorineural hearing loss (SNHL) and in children with “specific language impairment” (SLI), compared with a control group of children with normal development, to identify specific skill deficits. Three groups of preschool children aged between 2 and 6 years were assessed: 19 children with normal hearing, cognitive, and language development, 10 children with SNHL, and 20 children with SLI. The FE-PS 2-6 Battery was used for the assessment of preschool EFs, supplemented with the Modified Bell Test for the analysis of selective attention. Statistically significant differences were found between the two experimental groups and the control one, regarding the investigated skills. Children with SNHL showed a clear deficit in flexibility, whereas children with SLI had greater problems in self-regulation and management of waiting for gratification. Selective attention was found to be deficient in all three groups, with no statistically significant differences. This study shows that the skills investigated were found to be deficient in both SNHL and SLI patients. It is essential to start targeted exercises based on specific deficient skills as part of the rehabilitation program. It is of great importance to understand the consequences of EF deficit in preschool children to achieve an accurate diagnosis and carry out customized rehabilitation programs.

## 1. Introduction

In recent years, there has been an increased interest in the cognitive processes needed to develop skills to face a variety of situations and their dysfunction or disruption when central nervous system injury occurs. Specific cognitive resources are continuously modulated according to reach personal needs and goals. Executive functions (EFs) are essential to face new situations without patterns previously learned and are defined as “abilities that enable an individual to establish new patterns of behavior and ways of thinking and introspection of self”. The hypothesis is that EFs control the unfolding of cognitive processes rather than represent distinct cognitive operations [1,2].

The first documented evidence of the complexity of EFs and their neural basis is the famous case of Phineas Gage [3] and Lurija’s [4] observations on World War II veterans with neurosurgical consequences. The link between prefrontal area functions and motor behaviors, inhibition of immediate responses, abstraction, problem-solving, and reorientation according to the consequences of one’s actions was demonstrated. More recently, Grafman [5] suggested that the prefrontal cortex stores unique forms of hierarchical knowledge that appear as EF [3]. Language is also strongly correlated with EF. In the few existing longitudinal studies, it was found that early stages of language predict later self-regulatory abilities in children with typical development [6,7,8].

Hearing-impaired children with typical development can be considered a “pure condition” that is useful to explore the association between EFs and language, as their language development delay is due to the SNHL rather than a cognitive deficit. Indeed, some studies have observed poorer EF performances in children with SNHL and low scores on language tasks [5,9]. EF performances are also altered in children with SLI. This deficit could explain the absence of babbling between 5 and 10 months and the absence of deictic and referential gestures between 12 and 14 months, but few studies on this subject are available [10].

This study aims to evaluate EFs in children with SNHL and in patients with SLI. SNHL is a loss of function in the inner ear or its connections to the brain and affects more than 50 percent of children with hearing loss. Individuals with SNHL tend to show delays in language, deficits in executive functioning, and visual cognitive deficits, even following hearing amplification surgery and cochlear implants. This can occur for a variety of reasons, including delayed diagnosis or intervention, failed follow-ups, sporadic frequency of auditory–verbal therapy, and failure to use hearing aids. Language delay and cognitive dysfunction, which persist after intervention, are likely associated with altered brain structure and function in these patients. SLI represents a set of heterogeneous clinical pictures characterized by a disorder in language comprehension and production, starting from early developmental stages, with alterations of different types. These are children who, although they have no neurological, sensory, or relational problems, have limited vocabulary compared with their peers.

The goal is to investigate the skills and assess specific rehabilitation programs to help these patients achieve efficient mental processes and generate new flexible strategies to achieve goals and adapt to changes in everyday life, improve self-control and social interactions.

## 2. Materials and Methods

The test sample consisted of 49 subjects: 29 males (59.18%) and 20 females (40.82%) aged between 2 and 6 years old (average: 4.22 ± 1.14). Among the preschool children followed in the Centro di Audiofonologopedia in Rome, 20 children (13 males and 7 females, age range 4–6 years, mean age 5.6 ± 1.01 years) presented SLI (Specific Language Impairment, ICD-10, code F80.1), and 10 children (5 males and 5 females, age range 3–6 years, mean age 4.3 ± 0.96 years) presented profound (more than 85 dB HL) bilateral congenital SNHL (Sensorineural hearing loss bilateral, ICD-10, code H90.3): 6 children with hearing aids, 1 child with unilateral cochlear implant, and 3 children with bilateral cochlear implants.

○DIAGNOSTIC CRITERIASpecific Language Impairment: We included children with a present diagnosis of Language Disorder, made by standardized testing, who presented with reduced vocabulary, limited sentence structuring, language skills below those expected for age in comprehension and production, difficulties not associated with additional impairments (sensory impairments or intellectual disabilities), and IQ in the normal range, which was rated by Leiter-3 [11] or WISC-IV [12].Hypoacusis: We included children with a present diagnosis of Sensorineural Deafness with mixed speech disorder, bilateral sensorineural hearing aid, or bilateral or unilateral cochlear implant, who had IQs in the normal range.Control Group (normotypic children): We included children who had normal levels of verbal production and comprehension, with IQs in the normal range.

A control group of 19 children (13 males and 6 females, age range 2–5 years, mean age 3.6 ± 0.96 years) with normal hearing and language development were recruited from the “Gli amici di Tippete” nursery school in Ladispoli. All parents of the subjects who took part in the study signed an informed consent form. The children in each group underwent the assessment of EFs with the FE-PS 2-6 battery [13] and the assessment of selective attention skills with the Bells Test [14].

The project was approved by the Internal Board Review of the University of L’Aquila and was given the sequential identification number 13/2022 (approval date: 29 March 2022).

### 2.1. FE-PS 2-6 Battery

The FE-PS 2-6 battery (Edizioni Centro Studi Erickson, Trento, Italy) consists of 10 specific tests for preschool children aged from 2 to 6 years to evaluate EFs which are important in the development process of purposeful and complex behaviors to face new situations with a lack of knowledge and assessed automatisms. One of the EF functions is precisely to control cognitive resources when automatisms do not support those behaviors necessary to achieve individual goals. EFs are also involved in cognitive and behavioral self-regulation in preschool children and enable the coordination and modulation of cognitive and emotional processes and behavioral responses. These tests are based on well-established experimental paradigms on the development of EF in children with typical and atypical development. The ten tests of FE-PS 2-6 are useful to detect a dysfunctional EF development in children with language problems and attention and hyperactivity disorders. The FE-PS 2-6 mainly evaluates items such as inhibitory processes, response inhibition and interference management, delayed gratification, working memory, and flexibility [13].

### 2.2. Biancardi–Stoppa Modified Bell Test

The Biancardi–Stoppa modified Bells test was used to assess selective attention. This test is easy to understand and administer and allows the assessment of selectivity related to the recognition of a target stimulus by means of a focused attention task. It is, therefore, useful to assess the level of selective and sustained visual and visual–spatial attention in children aged between 4 and 14 years old [14].

Two types of scores are obtained from the administration of this test. The first one evaluates the rapidity and it’s based on the total number of bells detected in the first 30 s of search with the corresponding standard deviations. The second score evaluates the accuracy, and it represents the mean and standard deviation of the number of bells detected in the 120 s.

○STATISTICAL ANALYSIS

Once the three groups had been assessed using the EF-PS 2-6 battery and the Bell Test, the results obtained in the individual tests were stored on a magnetic stand, and statistical analyses were performed using Stata Statistical Release 12 software (Stata Corp LP, College Station, TX, USA) using the Kruskal–Wallis test (*p*-values < 0.05 were considered statistically significant). The characteristics of the study sample were analyzed using descriptive statistics and the quantitative variables were represented using mean and standard deviation (mean ± DS). The mean scores in the three groups (SNHL, SLI, and typical cognitive and language development) were compared using the Kruskal–Wallis test. Values of *p* < 0.05 were considered statistically significant. When the differences were found to be statistically significant, post hoc analysis for pairwise comparisons was possible by means of the Mann–Whitney test, while Bonferroni’s correction counteracted the problem of multiple comparisons (*p* < 0.0167).

## 3. Results

Age, gender, and results obtained from the standardized test of evaluation of language (TVL) [15] to assess the level of language development in its different components (production and comprehension) in the control group, SLI, and SNHL are reported in Table 1, Table 2 and Table 3. Hearing loss (in dB) and devices used by the subjects are also reported for the SNHL group in Table 3.

The following results were obtained with the FE-PS 2-6 battery and subdivided according to the individual cognitive abilities (Table 4). Inhibition was assessed by the “Draw a circle”, “Stroop day and night”, and “The elephant and the bear” tests. Children with SLI faced greater difficulties in the “Stroop day and night” and “The elephant and the bear” tests with inhibition of the dominant verbal response production and the prevalence of a nondominant one, resulting in inhibiting/activating motor responses following a verbal request. The Kruskal–Wallis test showed a statistically significant difference (*p* < 0.05) in these two tests. The post hoc analysis was performed and showed a greater difficulty in children with SLI compared with the control group (Table 5). Working memory was measured by means of the tests “Compare figures”, “The color and shape game” and “Keep in mind”. Statistical significance was shown in the first two tests for the SNHL and SLI groups, with greater difficulties if compared with the control group. A more pronounced deficit was found in SNHL children who received the lower score. No significant differences emerged in the “Keep in mind” test, but lower scores were found in children with SNHL. Flexibility was assessed by means of the tests “The fish game” and “The flower and star game”. In the first test, children with SLI scored lower than the other two groups without statistical significance. In the second test, children with SNHL scored lower than the other two groups without statistical significance. Emotional self-regulation was assessed with the “Wrapping the package” test and no significant differences were found between the experimental groups compared with the control group. In the “The gift” test, children with SLI experienced greater difficulties in self-regulation and in the expectation of gratification if compared with the control group.

Selective attention, assessed by means of the “Bell Test”, was found to be impaired in all three groups (Table 6) without statistically significant differences.

## 4. Discussion

EFs are related to a wide range of cognitive abilities mainly regulated by the prefrontal cortex, including inhibition, working memory, flexibility, emotional self-regulation, and attention [3]. EFs support learning processes and the ability to modify behavioral patterns and are of crucial importance in preschool children to develop new executive skills. Understanding EFs and the consequences of a deficit in preschool age is essential for diagnosis and assessing custom-made rehabilitation programs. The definition of EFs is complex and refers to a neuropsychological concept, which describes a set of abilities that enable individuals to organize and adapt their behaviors to future goals. In 2007, the World Health Organization and the American Occupational Therapy Association described EFs as “higher-level cognitive functions related to complex goal-oriented behavior in all domains of life”. In neuroscience, they are seen as fundamental building blocks for the development of cognitive and social skills [16]. It is important to know the developmental EFs’ process to understand their role in preschool children and adolescents. EFs appear in childhood in the first year of life and develop gradually throughout adolescence until early adulthood, together with the neurological functions of the prefrontal cortex. A decline associated with aging will follow. EF development differs from person to person, but the processing of new information, goal setting, and cognitive flexibility are relatively mature by the age of 12 years old [17]. EFs support learning processes and enable individuals to acquire knowledge and have effective social interactions, as they are considered the mechanism of all instrumental activities in daily life. They do not represent a single function but consist of several dissociable cognitive abilities that enable new behavioral patterns that are more efficient in coping with new and complex situations [16]. Working memory is a cognitive system that allows individuals to temporarily store and manipulate information to carry out complex tasks such as reasoning, comprehension, and learning. It refers to the mental capacity to elaborate the information that is no longer perceptually present, for the necessary time to mentally work on it. This cognitive task is useful for various purposes like problem-solving or decision making. Working memory is a key component of EFs and is a limited resource that can be easily overwhelmed by distractions or an excessive load of information [18,19]. The fundamental aspect of working memory is the decoding of information, paying attention to relevant ones, and replacing them when they are no longer necessary [1]. Inhibitory control is another essential EF, which involves the ability to actively control one’s attention, behavior, thoughts, and emotions in order to manage strong internal predispositions or external attractions. In other words, it is the ability to suppress thoughts that interfere with an appropriate response to the stimulus in favor of controlled responses and blocking automatic ones. It is supposed that the subthalamic nucleus plays a critical role in preventing such impulsive or premature responses [19]. Inhibition is closely related to working memory and interacts with it and cognitive control to modulate adaptive behavior. On the one hand, working memory supports inhibitory control because one must keep the goal in mind to know what is relevant or appropriate and what to inhibit. On the other hand, inhibitory control supports working memory: to relate different ideas or events, one must be able to focus on one thing at a time, as well as recombine ideas and facts in new and creative ways and avoid the recurrence of old patterns of thought [18,19]. The third fundamental EF is represented by cognitive flexibility, which is based on the previous ones and develops later. It is a distinct hallmark of human thought and allows one to quickly adapt to environmental changes and generate new ideas that promote growth and discovery. This adaptability allows the individual to be autonomous and to act through independent and intentional behavior. It enables one to change spatial or interpersonal perspectives, allowing different points of view and the inhibition of previous perspectives, as well as activating new ones in working memory. The appearance of environmental conditions or past experiences that interfere with the planning process carried out by the subject in any of these phases can lead to cognitive and behavioral rigidity. Children who have difficulties with this ability are not able to change their behavior in response to the context, and when solving a problem, they make perseverance errors or if they provide an incorrect answer, they continue to make mistakes, ignoring external feedback. Very often, these children have difficulties and slowness in analyzing a text or mathematical problems, so it is important to promote, especially in the early years of schooling, the ability to face a new situation and the tools to cope with it [20]. Attention is the ability to focus on certain elements, selecting what is useful, filtering it from all the information present, and simultaneously inhibiting distracting stimuli. The environment sends a multitude of visual, auditory, tactile, and olfactory information to the brain, but because our cognitive system has limited resources, only a portion of this incoming information is processed to become conscious: it is attention that determines which signals will arrive at the conscious experience. Experimental evidence has hypothesized that after a first phase in which all the information is analyzed and only the most relevant signals are recognized and selected, the other information remains available and can eventually be recalled in other situations. However, these theories cannot explain how some stimuli, even if irrelevant to the subject’s response, can still exceed the gain threshold for processing [20]. Planning is a higher-level cognitive function that includes the processes involved in formulating, evaluating, and selecting the actions necessary to achieve a goal, in other words, it consists of the ability to imagine how to reach a given goal [18,19]. For effective planning, it is necessary to anticipate the consequences of an action on others: in solving a task, one must be able to construct a mental map of the correct way through the anticipation of the problem solution in a functional manner. Planning is related to the abilities of abstraction, reasoning, and cognitive flexibility: to plan and properly solve a task one must be able to quickly switch from one concept to another and to know how to assign different meanings to the same concept. Furthermore, planning is involved in other higher cognitive processes such as problem-solving and decision making [20]. Emotional self-regulation is the ability to manage emotions and control feelings to maintain optimal levels of emotional, motivational, and cognitive excitement. This ability overlaps with inhibitory control: without inhibition, the individual is guided by impulses and environmental stimuli, but through self-regulation, the possibility of choice and reaction can be exercised [19].

In the literature, there are few studies that have used the FE-PS 2-6 evaluation battery to assess EFs in preschool children [13]. In this study, we made a comparison between children aged 2 to 6 years, divided into three groups: the first group consists of subjects with SNHL, the second group of subjects with SLI, and the third group of children with normal hearing, cognitive, and linguistic development. The goal was to evaluate the EFs in these groups of children, as recent studies [6,17,21,22,23,24,25,26,27] have revealed that subjects with SNHL and SLI are more deficient in executive abilities. Therefore, the EFs were evaluated using the FE-PS 2-6 battery, which allowed the study of individual cognitive functions using one or more trials. Inhibition, working memory, flexibility, and attention were examined. Selective attention was also evaluated using the “modified bells test” [14]. In this study, children with hearing loss encountered greater difficulties with inhibition, particularly in the ability to inhibit continuous motor response in the “Draw a circle” test. Significant difficulties were also found in working memory, where the SNHL group scored lower in all tests (“Compare figures”, “The color and shape game”, and “Keep in mind”). Subjects with SNHL have language delays, executive function, and visual cognitive deficits for different reasons, including delayed diagnosis or intervention, failed follow-up, sporadic auditory–verbal therapy, and nonuse of hearing aids. The language delay and cognitive dysfunction that persists after intervention are probably related to altered brain structure and function in these patients. Therefore, the study of brain changes is crucial to clarify the basic mechanisms of neuroplasticity that may explain abnormal brain function [28,29,30], and electroencephalographic recordings in children with hearing impairment have shown differences in the bilateral frontal cortex (closely related to executive abilities) and left temporal-frontal area (involved in expressive language) neural organization. A weaker development of these cortical areas may be reflected in poorer language and lower executive functions in deaf children. SHNL in children has a profound effect on communication and compared with normal-hearing peers, a higher risk of adverse social and emotional development, which may lead to significant behavioral problems [31]. Despite the successes of interventions with hearing aids or cochlear implants, up to 50% of children with hearing difficulties have behavioral problems, and it is precisely the presence of these behaviors that further complicates language development and social development [29,32]. Hearing deprivation in childhood can have cognitive effects that can extend beyond language abilities to more general areas. It has been observed that children with SNHL have problems performing a series of tasks that fall within the scope of executive functions [33]. Spoken language is facilitated by auditory experience, improves the development and use of executive functions, serves as a tool to control attention and behavior, assists working memory, and organizes complex information [22]. Early deprivation of auditory experience due to SNHL and the restoration of some components of hearing with a cochlear implant can influence executive function outcomes in preverbal deaf children who receive the implant [34,35,36,37].

The major difficulties for children with SLI were related to inhibition, as demonstrated by specific tests (“Stroop day and night” and “Bear and elephant”). Recent studies have investigated the role of executive functions in information processing in children with SLI, showing that these children have greater difficulties than their peers with typical language development in inhibition tasks, working memory, and attention control. These deficits are related to children’s linguistic competence. The literature on this subject is limited and presents methodological problems in several studies; furthermore, authors report different stimuli and procedures, which may explain inconsistent results [23]. Children with speech and language disorders face different linguistic challenges during childhood compared with their peers. With development, advanced reasoning and complex social interactions are required as well as the interpretation of secondary meanings. Adequate performances in inhibitory control and cognitive flexibility are requested at school age and must adapt to changing environments and the presence of more distractions and external pressures. Three key components of EFs related to speech and language disorders have been identified: updating information held in working memory, inhibiting unwanted responses and behaviors, and flexibility between mental tasks [38].

Studies on children with speech and language disorders have shown poorer working memory than children of the same age with typical language development, concluding that children with speech and language disorders perform worse than peers with typical language development on visual tasks that require both short-term memory and working memory. Thus, children with speech and language disorders appear to show working memory deficits in both visual and auditory modes. Most research also suggests that children with speech and language disorders show poor inhibition, which appears to affect both visual and auditory and verbal measures. However, these conclusions can be considered provisional due to a limited number of studies and inconsistencies regarding the presence or absence of inhibitory deficits in speech and language disorders, which can be explained by the speed of stimulus presentation, type, or clarity. Finally, there is a discrepancy between the performance of simple and complex tasks on measures of cognitive flexibility between children with speech and language disorders and those with typical language development. However, several studies indicate that this ability is not impaired in children with speech and language disorders, despite responding more slowly and making more errors than peers with typical language development in various tasks [25]. The present study revealed that both children with SNHL and SLI were lacking in flexibility and emotional self-regulation compared with the control group. Finally, attention was deficient in all the three groups examined. We may hypothesize that selective attention deficit in the control group was related to young children’s exposure to screen-media activities. While Samson et al. [39] found a significant positive association between time spent playing recreational videogames and selective attention, focusing on a tripartite model of attention, much screen-media literature has focused on potential negative impacts and associations with attention deficit hyperactivity [40,41,42]. Further studies are needed to investigate the impact of television and video game exposure on the visual sustained attention measure when the task is to point to targets among distractors on a paper sheet.

### Limits of the Study

The relatively small number of study participants raises concerns about the generalization of the results to a larger population. Additionally, a small study group is quite diverse in relation to children with hearing loss (e.g., in the case of a unilateral CI, the side may be important).

Our study did not analyze potential confounding variables, such as parental education level, that could potentially influence the results. Taking these factors into account in future research could provide a more complete understanding of the examined relationships.

Another limitation of this study is the lack of long-term follow-up. This study concerned children aged 2 to 6 years, and this is the period when the functions in question develop and this development, even in children without deficiencies, may proceed at a different pace.

Although our study highlights the importance of further research on selective attention, the lack of in-depth analysis of this aspect may make it difficult to fully understand its implications for executive function disorders in children.

## 5. Conclusions

In conclusion, the present study showed that children with SNHL were found to have greater deficits in flexibility and working memory, whereas children with SLI had greater difficulties in the ability to self-regulate and wait for gratification. Selective attention was deficient in all three groups, with no statistically significant differences. Consequently, in children with hearing loss, it is essential to introduce exercises to support and increment executive skills, in particular flexibility and working memory, as part of the rehabilitation program from an early age. In children with SLI, deficiencies in inhibition make it necessary to have an early intervention focused on this ability. It would also be essential to conduct more in-depth studies on selective attention, as it was found to be lacking in all the groups studied.

The FE-PS 2-6 battery deserves a final consideration, since it has proven to be a usable tool in school contexts, as it was possible to perform all the tests during school hours without compromising the normal activities of the children. In addition, the variety of tests available allows one to choose the activities based on the preferences and performance of the child, thus avoiding frustration, and maintaining high motivation. The continuation of the present study requires an increase in the number of participants in the experimental groups, as well as a follow-up to a targeted intervention on the skills investigated.

## Figures and Tables

**Table 1 brainsci-14-00491-t001:** Age, gender, production, and comprehension in the control group. SD: standard deviation.

Patient	Age (Years)	Gender	PRODUCTION	COMPREHENSION
1	4	M	1 SD	medium
2	5	M	2 SD	good
3	5	F	2 SD	good
4	3	F	>2 SD	very good
5	4	M	2 SD	medium-high
6	4	M	2 SD	medium-high
7	3	M	>2 SD	good
8	3	F	1 SD	medium
9	3	F	1 SD	medium
10	2	F	2 SD	medium-high
11	3	M	2 SD	medium-high
12	3	M	1 SD	medium
13	4	M	1 SD	medium-high
14	3	M	1 SD	good
15	5	M	2 SD	medium-high
16	2	M	2 SD	medium-high
17	3	M	1 SD	medium
18	4	F	1 SD	medium
19	5	M	2 SD	good

**Table 2 brainsci-14-00491-t002:** Age, gender, production, and comprehension in the SLI group. SD: standard deviation.

Patient	Age (Years)	Gender	PRODUCTION	COMPREHENSION
1	4	M	<1 SD	medium-low
2	6	M	<2 SD	low
3	5	M	<2 SD	medium-low
4	6	F	<2 SD	medium
5	4	M	<2 SD	medium
6	4	M	<1 SD	medium
7	5	F	<1 SD	medium-low
8	6	F	<2 SD	medium
9	5	M	<1 SD	medium
10	5	M	<2 SD	medium-low
11	4	M	<1 SD	medium
12	4	F	<1 SD	medium
13	4	F	<1 SD	medium-low
14	5	M	<2 SD	medium
15	5	M	<1 SD	medium
16	6	M	<1 SD	medium-low
17	5	F	<1 SD	medium
18	4	F	<2 SD	medium-low
19	5	M	<1 SD	medium-low
20	6	F	<1 SD	medium

**Table 3 brainsci-14-00491-t003:** Age, gender, medium hearing loss in dB, prosthetic devices (hearing aids HA; cochlear implants CI), production, and comprehension in the SNHL group. SD: standard deviation.

Patient	Age (Years)	Gender	Hearing Loss	Prosthetic Devices	PRODUCTION	COMPREHENSION
1I	4	M	95 dB	Bilateral retroauricolar HA	<1 SD	insufficient
2I	3	F	110 dB	Bilateral retroauricolar HA	<2 SD	medium-low
3I	4	F	100 dB	bilateral CI	<1 SD	medium
4I	6	M	110 dB	bilateral CI	<2 SD	medium
5I	4	M	95 dB	Bilateral retroauricolar HA	<2 SD	medium-low
6I	2	F	110 dB	Bilateral retroauricolar HA	<2 SD	insufficient
7I	4	M	100 dB	Bilateral retroauricolar HA	<1 SD	medium
8I	4	F	100 dB	Bilateral retroauricolar HA	<2 SD	medium-low
9I	3	M	95 dB	bilateral CI	<1 SD	medium
10I	6	F	115 dB	right unilateral CI	<1 SD	medium

**Table 4 brainsci-14-00491-t004:** Quantitative variables with mean and standard deviation.

	Control N = 19	SNHL N = 10	SLI N = 20	*p*-Value
Draw a circle				
Percentile	42.63 ± 28.49	32.50 ± 31.51	40 ± 30.35	0.6226
z-score	−0.19 ± 0.85	−0.42 ± 1.16	−0.56 ± 1.42	0.7968
Stroop day and night				
Percentile	75.33 ± 25.81	71.25 ± 31.98	36 ± 30.07	0.0046
z-score	0.85 ± 0.99	0.66 ± 1.15	−0.52 ± 1.20	0.0007
The elephant and the bear				
Percentile	80.39 ± 14.93	58.33 ± 42.52	57 ± 35.29	0.3052
z-score	1.01 ± 0.51	0.07 ± 1.05	−0.14 ± 0.99	0.0207
Compare figures				
Percentile	77.08 ± 25.27	41.88 ± 30.82	48 ± 30.63	0.0057
z-score	0.91 ± 1.05	−0.54 ± 1.80	−0.16 ± 1.34	0.0616
The fish game				
Percentile	52.37 ± 25.13	53.13 ± 29.87	39.5 ± 27.24	0.2379
z-score	0.19 ± 0.67	0.34 ± 1.23	−0.57 ± 1.97	0.7210
Wrapping the package				
Percentile	68.13 ± 26.83	47.5 ± 39.59	51 ± 38.18	0.2074
z-score	0.38 ± 0.49	−0.37 ± 1.22	−0.69 ± 2.11	0.2536
The gift				
Percentile	95 ± 0	75 ± 28.66	86.75 ± 12.06	0.0069
z-score	16.45 ± 11.11	7.94 ± 9.65	4.93 ± 5.61	0.0026
The color and shape game				
Percentile	92.19 ± 2.56	50 ± 13.36	62.5 ± 29.49	0.0002
z-score	1.07 ± 0.20	0.27 ± 0.21	0.44 ± 0.78	0.0008
Keep in mind				
Percentile	85 ± 14.14	48 ± 38.18	55 ± 31.23	0.3573
z-score	1.3 ± 0.58	−0.09 ± 1.23	0.22 ± 0.98	0.2906
The flower and star game				
Percentile	53.3 ± 26.46	20 ± 18.37	47.3 ± 27.98	0.0638
z-score	0.15 ± 0.73	−2.84 ± 4.15	0.01 ± 1.27	0.0385
The bell test				
z-score speed	−1.35 ± 0.90	−1.20 ± 1.37	−0.89 ± 1.32	0.6792
z-score accuracy	−1.35 ± 1.12	−0.60 ± 0.67	−1.19 ± 1.53	0.2860

**Table 5 brainsci-14-00491-t005:** Post hoc analysis pairwise comparisons for variables with *p* < 0.05.

Post Hoc Analysis		
	Control vs. SNHL	Control vs. SLI
Stroop day and night (percentile)	0.8767	0.0017
Stroop day and night (z-score)	0.5144	0.0002
The elephant and the bear (z-score)	0.0895	0.0002
Compare figures (percentile)	0.0097	0.0002
The gift (percentile)	0.0105	0.0012
The gift (z-score)	0.0707	0.0004
The color and shape game (percentile)	0.0000	0.0035
The color and shape game (z-score)	0.0001	0.0122
The flower and star game (z-score)	0.0195	0.4224

Using ranksum test with Bonferroni adjustment for multiple comparisons, *p* < 0.0083 comparisons.

**Table 6 brainsci-14-00491-t006:** Quantitative variables Bell Test.

	Control N = 19	SNHL N = 10	SLI N = 20	*p*-Value
The Bell test				
z-score speed	−1.35 ± 0.9004073	−1.20167 ± 1.37367	−0.89 ± 1.323631	0.6792
z-score accuracy	−1.35429 ± 1.118702	−0.60167 ± 0.67185	−1.18875 ± 1.53144	0.2860

## Data Availability

Data are contained within the article.

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
