# Peer review of "Assessment of Executive Functions in Children with Sensorineural Hearing Loss and in Children with Specific Language Impairment: Preliminary Reports"

_brainsci, 2024, doi:10.3390/brainsci14050491_

Round 1

Reviewer 1 Report

Comments and Suggestions for Authors

Dear Authors
I recently had the opportunity to read your research. While I found the study to be insightful and well-conducted in many respects, I would like to draw your attention to some areas that require further consideration and improvement.

1.       Small Study Sample: While I appreciate your appreciation of the need for additional research to confirm the results, the relatively small number of study participants raises concerns about the generalizability of the results to a larger population. Additionally, with a small study group, it is quite diverse in relation to children with hearing loss (e.g. in the case of a unilateral implant, the side may be important).

2.       No analysis of confounding factors: It appears that your study did not analyze potential confounding variables, such as parental education level, that could potentially influence the results. Taking these factors into account in future research could provide a more complete understanding of the examined relationships.

3.       Lack of long-term follow-up: The study concerned children aged 2 to 6 years, and this is the period when the functions in question develop and this development, even in children without deficiencies, may proceed at a different pace.

4.       Limited conclusions regarding selective attention: Although your study highlights the importance of further research on selective attention, the lack of in-depth analysis of this aspect may make it difficult to fully understand its implications for executive function disorders in children. Delving into this area can enrich your findings and contribute to the existing literature.

     Overall, I congratulate you on a thoroughly conducted study. I believe that considering these issues can further increase the impact and importance of your work. I look forward to seeing your research progress in the future.

Best Regards

Author Response

Many thanks for your appreciation and time spent reviewing the paper.

We have added your suggestions about small sample size, no analysis of confounding factors, lack of long-term follow-up and limited conclusions regarding selective attention as limits of the study at the end of the discussion.

Reviewer 2 Report

Comments and Suggestions for Authors

I would like to thank the authors for submitting for review a very interesting article that raises important issues in the development of children with hearing disabilities. The authors of the paper, in the conclusion, point out the small study group and the need to conduct research on a larger group of people, which is indeed a weak element of the paper, maybe, in view of this, modify the title by adding "preliminary reports"?

Only 24% of the references cited are from the last 5 years. This is all the more surprising because in recent years there has been an increase in the literature on the analysis of executive functioning in children receiving cochlear implants. It would be worthwhile to cite these papers in the discussion and relate them to your own results.

I propose to standardize the way of presenting data, if for one group the mean age with standard deviation is given, it would be worthwhile to present the subsequent ones similarly.
It is worth summarizing Tables 1 and 2 (both tables are not very understandable) in a descriptive way, as well as indicating whether the analyzed groups were statistically different from each other in terms of age and gender.

I would more clearly describe the patient groups that are later described as SLI and SNHL in Materials and Methods.

In the inclusion criteria, the authors use the term "IQ in the normal range" - how was this analyzed?

A revision of punctuation is needed.

Author Response

Many thanks for the time spent reviewing this paper and for your appreciation and suggestions.

I would like to thank the authors for submitting for review a very interesting article that raises important issues in the development of children with hearing disabilities. The authors of the paper, in the conclusion, point out the small study group and the need to conduct research on a larger group of people, which is indeed a weak element of the paper, maybe, in view of this, modify the title by adding "preliminary reports"?

We have modified the title adding "preliminary reports" as you suggested.

Only 24% of the references cited are from the last 5 years. This is all the more surprising because in recent years there has been an increase in the literature on the analysis of executive functioning in children receiving cochlear implants. It would be worthwhile to cite these papers in the discussion and relate them to your own results.

We have added 13 references more recent as you suggested.

I propose to standardize the way of presenting data, if for one group the mean age with standard deviation is given, it would be worthwhile to present the subsequent ones similarly.

We have added standard deviation of age for every group (SLI, SNHL and control) as you suggested.
It is worth summarizing Tables 1 and 2 (both tables are not very understandable) in a descriptive way, as well as indicating whether the analyzed groups were statistically different from each other in terms of age and gender.

We have added the informations of the table 1 and 2 in the text in a descriptive way.

I would more clearly describe the patient groups that are later described as SLI and SNHL in Materials and Methods.

We have added a section on SLI and SNHL in the introduction as requested.

In the inclusion criteria, the authors use the term "IQ in the normal range" - how was this analyzed?

We have added in the text that IQ in the normal range 

I would like to thank the authors for submitting for review a very interesting article that raises important issues in the development of children with hearing disabilities. The authors of the paper, in the conclusion, point out the small study group and the need to conduct research on a larger group of people, which is indeed a weak element of the paper, maybe, in view of this, modify the title by adding "preliminary reports"?

Only 24% of the references cited are from the last 5 years. This is all the more surprising because in recent years there has been an increase in the literature on the analysis of executive functioning in children receiving cochlear implants. It would be worthwhile to cite these papers in the discussion and relate them to your own results.

I propose to standardize the way of presenting data, if for one group the mean age with standard deviation is given, it would be worthwhile to present the subsequent ones similarly.
It is worth summarizing Tables 1 and 2 (both tables are not very understandable) in a descriptive way, as well as indicating whether the analyzed groups were statistically different from each other in terms of age and gender.

I would more clearly describe the patient groups that are later described as SLI and SNHL in Materials and Methods.

In the inclusion criteria, the authors use the term "IQ in the normal range" - how was this analyzed?

We have added in the text that IQ in the normal range was rated by Leiter 3 or WISC-IV

A revision of punctuation is needed.

We have made a revision of punctuation as requested.

Reviewer 3 Report

Comments and Suggestions for Authors

It is an interesting research regarding executive function in children with different disabilities (hearing loss and language impairment).  It is an original research, there are previous publication regarding this area but not in groups like this in preschool children.  

Abstract is well written and give us good preview of the reserch. 

There are few things authors shold try to explain and to give readers information regarding possible shortage  of this research (in the discussion and conclusion).

Children in this research have between 2 - 6 years this is a very large range for this age, authors should explain if the difference in age could influence there results. 

Children with hearing loss (only 10 children) is very heterogeneous- some with hearing aid other with cochlear implant - this also needs to be explained, research would be much better if the children have similar hearing loss – and treatment modality. 

Authors should better explain statement form introduction and conclusion - Selective attention was  deficient in all the three groups - How do they explain that selective attention was deficient in control group?

Table 3. - Bilateral retrocochlear HA – not sure what is this probably hearing aid or retraouricular hearing aid – it should be corrected.

Table 5 it should be explained what is group A, B and C. 

References seem to be correct but this is international journal, seven references in italian language is too much, authors should consider to have more references in english language for the english speaking readers. 

Author Response

Many thanks for your suggestions and time spent reviewing the paper.

Children in this research have between 2 - 6 years this is a very large range for this age, authors should explain if the difference in age could influence there results. 

The difference in age could not influence the results because the test used for the executive function is standardized for pre-school children from 2 to 6 years.

Children with hearing loss (only 10 children) is very heterogeneous- some with hearing aid other with cochlear implant - this also needs to be explained, research would be much better if the children have similar hearing loss – and treatment modality. 

The treatment modality and the tests used for the language are similar also for different kind of hearing loss because they work for hearing perception.

Authors should better explain statement form introduction and conclusion - Selective attention was  deficient in all the three groups - How do they explain that selective attention was deficient in control group?

We may hypothesize that selective attention deficit in the control group was related to young children’s exposure to screen-media activities.

While Samson et al. (Samson AD, Rohr CS, Park S, Arora A, Ip A, Tansey R, Comessotti T, Madigan S, Dewey D, Bray S. Videogame exposure positively associates with selective attention in a cross-sectional sample of young children. PLos One 2021; 16(9): e0257877.)  found a significant positive association between time spent playing recreational videogames and selective attention, focusing on a tripartite model of attention, much screen-media literature has focused on potential negative impacts and associations with attention deficit hyperactivity (Montagni I, Guichard E, Kurth T. Association of screen time with self-perceived attention problems and hyperactivity levels in French students: a cross-sectional study. BMJ Open. 2016Feb26;6(2):e009089. Tamana SK, Ezeugwu V, Chikuma J, Lefebvre DL, Azad MB, Moraes TJ, et al. Screen-time is associated with inattention problems in preschoolers: Results from the CHILD birth cohort study. PLoS One. 2019;14: e0213995. Swing EL, Gentile DA, Anderson CA, Walsh DA. Television and video game exposure and the development of attention problems. Pediatrics. 2010Jul05;126: 214–221.) Further studies are needed to investigate the impact of television and video game exposure on the visual sustained attention measure when the task is to point to targets among distractors on a paper sheet. 

Table 3. - Bilateral retrocochlear HA – not sure what is this probably hearing aid or retraouricular hearing aid – it should be corrected.

Yes, we have corrected with bilateral retroauricolar HA

Table 5 it should be explained what is group A, B and C. 

We have corrected group A, B and C with Control, SNHL and SLI.

References seem to be correct but this is international journal, seven references in italian language is too much, authors should consider to have more references in english language for the english speaking readers. 

We have added 13 international references more recent.

Round 2

Reviewer 2 Report

Comments and Suggestions for Authors

Thank you for sending the corrected text. I agree with the corrections made and accept the authors' response.